# Exploring the causes of COPD misdiagnosis in primary care: A mixed methods study

**Ketan Patel**[1]*, **Daniel J. Smith**[2], **Christopher C. Huntley**[1], **Sunita D. Channa**[3], **Anita Pye**[3], **Andrew P. Dickens**[4], **Nicola Gale**[5], **Alice M. Turner**[3]

1 University Hospitals Birmingham NHS Foundation Trust, Birmingham, United Kingdom, 2 University of Birmingham, Birmingham, United Kingdom, 3 Institute of Applied Health Research, University of Birmingham, Birmingham, United Kingdom, 4 Observational and Pragmatic Research Institute, Midview City, Singapore, 5 School of Social Policy, University of Birmingham, Birmingham, United Kingdom

* Ketan.patel1@nhs.net

**Data Availability Statement:** All relevant data are within the manuscript and its Supporting Information files.

## Abstract

### Background

Within primary care there exists a cohort of patients misdiagnosed with Chronic Obstructive Pulmonary Disease (COPD). Misdiagnosis can have a detrimental impact on healthcare finances and patient health and so understanding the factors leading to misdiagnosis is crucial in order to reduce misdiagnosis in the future. The objective of this study is to understand and explore the perceived causes of COPD misdiagnosis in primary care.

### Methods

A sequential mixed methods study, quantifying prevalence and features of patients misdiagnosed with COPD in primary care followed by a qualitative analysis to explore perceived causes of misdiagnosis. Quantitative data was collected for 206 patients identified as misdiagnosed with COPD within the INTEGR COPD study (NCT03482700). Qualitative data collected from 21 healthcare professionals involved in providing COPD care and 8 misdiagnosed patients who were recruited using a maximum variation purposive sampling.

### Results

Misinterpretation of spirometry results was the prevailing factor leading to patients initially being misdiagnosed with COPD, affecting 59% of misdiagnosed patients in this cohort. Of the 99 patients who were investigated for their underlying diagnosis; 41% had normal spirometry and 40% had asthma. Further investigation through qualitative methodology uncovered reluctance to challenge historical misdiagnoses and challenges in differential diagnosis as the underlying explanations for COPD misdiagnosis in this cohort.

### Conclusions

Patients historically diagnosed with COPD without spirometric evidence are at risk of remaining labelled and treated for COPD despite non-obstructive respiratory physiology, leading to a persistent cohort of patients misdiagnosed with COPD in primary care. The lack of spirometry services during and after the COVID19 pandemic in primary care risks adding

**Funding:** The INTEGR COPD study was funded by a non-commercial grant awarded by AstraZeneca (ESR-16-12347) following a peer-review process and sponsored by University Hospitals Birmingham NHS Foundation Trust (UHB). The funders had no role in study design, data collection and analysis, decision to publish, or preparation of the manuscript.

**Competing interests:** The authors have read the journal's policy and have the following competing interests: AMT and KP were supported financially in relation to this study through a non-commercial grant from AstraZeneca. AMT has received grants not in relation to this study from Chiesi, NIHR ARC & PSRC, CSL Behring and Grifols Biotherapeutics. DJS, CCH, SDC, AP, APD and NG declare no support from any organisation for the submitted work; no financial relationships with any organisations that might have an interest in the submitted work in the previous three years, and no other relationships or activities that could appear to have influenced the submitted work. This does not alter our adherence to PLOS ONE policies on sharing data and materials.

to the cohort of misdiagnosed patients. Support from respiratory specialists can potentially help to reduce the prevalence of COPD misdiagnosis in primary care.

## Trial registration

NCT03482700.

## Introduction

Chronic Obstructive Pulmonary Disease (COPD) in the United Kingdom (UK) is primarily diagnosed and managed by generalists in primary care [1]. Specialists are only involved in the care of patients when there is uncertainty with diagnosis or difficulty with management [2]. However, specialist involvement requires General Practitioners (GP) to identify the problem first, then make a referral [3].

Diagnosing COPD requires a combination of spirometry and clinical skills [4]. The ability to perform and interpret spirometric tests is necessary to confirm the presence of obstructive airways [5]. Primary care clinicians also need to be able to assess the clinical history and examination to determine if patients truly have COPD or other diseases that can mimic COPD symptoms [6]. These competencies require specialist training as well as constant maintenance to prevent deskilling [7].

Misdiagnosis leads to inappropriate treatment, which comes at a financial cost. The mainstay of treatment for COPD remains inhaled therapies [8]. COPD is estimated to cost the National Health Service (NHS) £1.9 billion per year [9]. The proportion of COPD expenditure in the UK attributable to pharmacological therapies is uncertain, however, we do know that the European Respiratory Society (ERS) estimates that 30% of the direct costs of COPD in Europe are due to pharmacological management [10]. Inappropriate treatment can also lead to patient harm. The action of treating misdiagnosed patients with inhaled therapies can potentially lead to harm through increased risk of pneumonia due to inhaled steroids [11], as well as harm through failure to treat the true underlying illness [7]. The impact of delayed treatment of the true illness causing symptoms can be devastating for patients as it can impact their quality of life and socio-economic wellbeing [7].

Current literature suggests that difficulty with spirometry and differentiating COPD from asthma are the main causes of misdiagnosis in primary care [12]. There are no published qualitative studies exploring the perceived direct or contextual causes of COPD misdiagnosis in primary care.

The INTEGR COPD study aimed to explore the impact of integrating respiratory specialists into GP practices, and whether this could address these causes of misdiagnosis and reduce the burden of COPD misdiagnosis in primary care. The study involved reviewing the clinical evidence supporting the diagnosis of COPD in patients labelled as such, which resulted in the identification of a cohort of misdiagnosed patients. Utilising this cohort of misdiagnosed patients within the INTEGR COPD study, we have set out to understand why patients are misdiagnosed with COPD within primary care using a mixed methods approach.

## Methods

This study utilised a sequential, mixed methods explanatory approach [13]. Factors associated with COPD misdiagnosis were identified through collaborative review of misdiagnosed patients' clinical notes by the lead researcher (KP) and the patient's primary care clinician. These factors were quantified and were used to develop the topic guide for qualitative data

collection. Qualitative data was collected using semi-structured interviews with Health Care Professionals (HCP) and misdiagnosed patients to explore and understand why misdiagnosis occurred. A trial steering committee consisting of clinicians, academics and patient representatives were consulted throughout the study period, in particular their feedback was used to aid in the development of the interview topic guide.

## Participants

Patients were defined as misdiagnosed with COPD if they had an existing COPD label within their medical record without meeting clinical and spirometric diagnostic criteria set out by the Global Initiative for Chronic Obstructive Lung Diseases (GOLD) [4]. The decision to label a patient as misdiagnosed required agreement of the multi-disciplinary team (MDT), which consisted of respiratory physicians, respiratory nurses, GPs and primary care nurses. Misdiagnosed patients and HCPs who had been involved in providing COPD care were approached for participation in the qualitative component of this study through either email or telephone call and were informed of the purpose of the interview and the aims of the study. Informed written consent was obtained by those who agreed to participate. To ensure heterogeneity in the sample, maximum variation purposive sampling was used to select patients and HCPs for interview. The end point of recruitment to the qualitative component was determined by reaching theoretical saturation, whereby no new viewpoints or concepts were arising from the interviews [14, 15].

## Data collection

The cause of misdiagnosis was recorded for each patient, following a collaborative review of the patient's clinical records by the lead researcher (KP) and the patient's GP or primary care nurse, with disagreements resolved by a third clinician. Causes of misdiagnosis were described in narrative format for each patient initially during collaborative review of the clinical notes. The descriptions were then standardised into coded terms by the lead researcher (KP) and a clinical research fellow (CCH).

The underlying diagnosis, where available, was also recorded for patients who were part of the intervention arm of INTEGR COPD, as this data was recorded by the specialist during clinical review. The underlying diagnosis was based on the results of further clinical investigations where appropriate. New underlying diagnoses could not be recorded in the patients who were part of the control arm as additional clinical review to determine underlying pathology was not within the scope of the INTEGR COPD study.

Semi-structured interviews were conducted between March 2020 and March 2021 by lead researcher (KP) and medical student (DJS), both of whom had been trained in qualitative research methodology. Participants (patients and HCPs) were offered the opportunity to have their interview either face to face or via telephone. Participants were aware that DJS was a medical student and that KP was a respiratory physician. Interviews were audio recorded and transcribed.

The topics chosen for the topic guide (S1 File) were based on the findings from the quantitative analysis in this mixed methods study as well as current literature that explores the causes of misdiagnosis [12, 16] and barriers to diagnosing COPD adequately [17, 18].

## Analysis

Demographic characteristics were tabulated and descriptive statistics were used to determine the commonest identified factors leading to misdiagnosis of COPD. This information was then used to guide participant selection and develop the topic guide.

Interview transcripts were coded using Nvivo 14 and analysed using the Framework method for thematic analysis of qualitative data outlined by Gale et al. [19]. Three transcripts deemed to be rich and informative were selected to be coded and discussed by four researchers with mixed professional backgrounds, including clinical and non-clinical researchers. This process was repeated with a further three transcripts to identify additional codes after which an initial coding framework was developed, which is available in S2 File. The remaining transcripts were coded using this framework. Once all transcripts were coded, a matrix framework was developed with summarized data from the transcripts tabulated with their associated codes. Similarities between codes were identified and grouped to form categories, which were summarised into analytical memos, with illustrative quotes, which can be found in S3 File. Following discussion between the team, linkages between these categories were reviewed to form themes, this is illustrated in a thematic map available in S4 File.

### Ethical approval

The INTEGR COPD study was approved by West Midlands South Birmingham Research Ethics Committee (REC 17/WM/0342) and was registered on the clinicaltrials.gov database (NCT03482700). An amendment was made and approved to allow patients and HCPs to participate in semi-structured interviews to explore causes of COPD misdiagnosis.

### Results

Between December 2017 and May 2020, 1,458 patients had undergone diagnostic review as part of the INTEGR COPD study, of whom 206 (14%) were identified as having been misdiagnosed with COPD. The demographic and clinical characteristics of the misdiagnosed cohort are presented in Table 1. As expected, the mean FEV1/FVC ratio was greater than 0.70. Most patients were either ex-smokers or never smokers and were symptomatic of dyspnoea with an MRC score greater than 1. Majority of patients were from a deprived socioeconomic background, which was expected as the study recruited patients from an economically deprived region.

59% of patients were misdiagnosed with COPD due to difficulty interpreting the spirometry report and a further 18% were misdiagnosed due to the absence of spirometry (Table 2). Difficulties with spirometry were by far the commonest cause of misdiagnosis, however, it is important to note that 17% of patients were misdiagnosed due to misinterpretation of the clinical history.

Normal spirometry was the prevailing clinical finding within the cohort of misdiagnosed patients (Table 3). 40% of patients were found to have significant reversibility on spirometry and were subsequently diagnosed with asthma rather than COPD. Interestingly, of the 40 patients found to have asthma rather than COPD, 24 (60%) had a concurrent diagnosis of asthma already. Overall, 21% of misdiagnosed patients had a concurrent diagnosis of asthma within their electronic patient records. This indicates that there is an element of difficulty deciding between a diagnosis of COPD and asthma within this cohort.

The quantitative aspect of this study suggested that difficulty with spirometry in primary care was the root cause of COPD misdiagnosis, however, difficulty differentiating COPD from asthma also played a role. Based on these findings, topics surrounding "spirometry in primary care", "diagnosing COPD" and "differentiating COPD from asthma" were included in the topic guide.

23 HCPs and 12 misdiagnosed patients were approached for interviews, of which 2 HCPs and 4 misdiagnosed patients declined to participate, with no reasons given. 21 HCPs and 8 misdiagnosed patients were interviewed. A summary of their characteristics is presented in

**Table 1. Demographic and clinical characteristics for misdiagnosed patients in the intervention and control arms of INTEGR COPD.** Data presented as frequency (%) or mean (SD). FEV$_1$: Forced Expiratory Volume in 1 second; SD: Standard deviation.

| | Intervention | Control | Whole cohort |
|---|---|---|---|
| Number undergoing diagnostic review | 695 | 763 | 1458 |
| Number misdiagnosed (%) | 99 (14%) | 107 (14%) | 206 (14%) |
| *Demographics* | | | |
| Male (%) | 52 (53%) | 46 (43%) | 98 (48%) |
| Mean Age (SD) | 66 (13.78) | 65 (13.31) | 66 (13.54) |
| *Spirometry* | | | |
| Mean FEV1%predicted (SD) | 82.88% (17.72) | 78.10% (16.02) | 80.87% (17.19) |
| Mean FEV1/FVC (SD) | 0.77 (0.08) | 0.76 (0.09) | 0.76 (0.09) |
| *MRC Dyspnoea Score* | | | |
| MRC1 (%) | 24 (24%) | 14 (13%) | 38 (18%) |
| MRC2 (%) | 28 (28%) | 37 (35%) | 65 (32%) |
| MRC3 (%) | 17 (17%) | 29 (27%) | 46 (22%) |
| MRC4 (%) | 18 (18%) | 12 (11%) | 30 (15%) |
| MRC5 (%) | 3 (3%) | 1 (1%) | 4 (2%) |
| No MRC score recorded (%) | 9 (9%) | 14 (13%) | 23 (11%) |
| *Smoking Status* | | | |
| Smoker (%) | 33 (33%) | 42 (39%) | 75 (36%) |
| Ex-Smoker (%) | 50 (50%) | 45 (42%) | 95 (47%) |
| Never Smoker (%) | 14 (14%) | 19 (18%) | 33 (16%) |
| No smoking status (%) | 2 (2%) | 1 (1%) | 3 (1%) |
| *Co-morbidities* | | | |
| Asthma | 11 (11%) | 33 (31%) | 44 (21%) |
| *Index of Multiple Deprivation (IMD) decile* | | | |
| IMD 1st Decile (%)**(%)** | 54 (55%) | 64 (60%) | 118 (57%) |
| IMD 2nd Decile (%) | 31 (31%) | 21 (20%) | 52 (25%) |
| IMD 3rd Decile (%) | 7 (7%) | 10 (9%) | 17 (8%) |
| IMD 4th Decile (%) | 3 (3%) | 3 (3%) | 6 (3%) |
| IMD 5th Decile (%) | 1 (1%) | 6 (6%) | 7 (3%) |
| IMD 6th Decile (%) | 0 | 1 (1%) | 1 (0.5%) |
| IMD 7th Decile (%) | 0 | 0 | 0 |
| IMD 8th Decile (%) | 1 (1%) | 0 | 1 (0.5%) |
| IMD 9th Decile (%) | 1 (1%) | 1 (1%) | 2 (1%) |
| IMD 10th Decile (%) | 0 | 1 (1%) | 1 (0.5%) |
| Unknown (%) | 1 (1%) | 0 | 1 (0.5%) |

**Table 2. Identified causes of misdiagnosis within the misdiagnosed cohort of INTEGR COPD.** Data presented as frequency (%).

| Identified cause of misdiagnosis | Misdiagnosed cohort n = 206 |
|---|---|
| Misinterpreted spirometry | 122 (59%) |
| Spirometry not used | 37 (18%) |
| Misinterpreted clinical History | 34 (17%) |
| Poor quality spirometry | 10 (5%) |
| Miscommunication from secondary care | 2 (1%) |
| Unknown | 1 (<1%) |

**Table 3. Identified underlying diagnoses in misdiagnosed cohort within the intervention arm of INTEGR COPD.** Data presented as frequency (%).

| Underlying diagnosis | Misdiagnosed cohort–Intervention arm (n = 99) |
|---|---|
| Normal Spirometry | 41 (41%) |
| Asthma | 40 (40%) |
| Interstitial lung disease | 5 (5%) |
| Obesity | 4 (4%) |
| Restrictive disease | 4 (4%) |
| Heart failure | 2 (2%) |
| Bronchiectasis | 2 (2%) |
| Lung cancer | 1 (1%) |

Table 4. Two likely explanations for misdiagnosis were identified: *Reluctance to challenge historical misdiagnosis* and *Challenges in differential diagnosis.*

## Explanation 1—Reluctance to challenge historical misdiagnosis

Patients found to have been misdiagnosed were perceived to be patients that had historical misdiagnosis due to limitations in the diagnostic process. Participants focused on the limited availability of spirometry to confirm diagnoses of COPD, thus many historical diagnoses were made based on clinical findings alone:

> "Okay, some of them may have been historically misdiagnosed. We only started doing spirometry about five years ago, I think or something like this. I'm certain about it. Because I was diagnosing COPD on history in the past, smoker who's got recurrent infections, wheezing etc and lots of things over there. And I'm talking about more than 10 years ago, or even 15 years ago, spirometry started coming in."–GP1

A historical diagnosis of COPD was perceived to lead clinicians down a single pathway and prevent them from considering alternative diagnoses even if the original COPD diagnosis was made without spirometry:

> "I've come across a few people who have just been diagnosed with COPD, but have actually had that overlap with asthma. And I find once someone is diagnosed as COPD, and we sort of forget that there can be other things"–GP2

Participants appreciated that spirometry played a role in the diagnostic process and was also perceived as a useful tool for annual monitoring of patients with COPD in primary care:

**Table 4. Summary of participant characteristics.** Data presented as frequency and mean values with range given in brackets.

| Participant Role | Number of participants | Number of male participants | Mean age | Mean years of clinical practice |
|---|---|---|---|---|
| Patient | 8 | 4 | 66 (54–79) | N/A |
| General practitioner | 12 | 6 | 50 (30–75) | 20 (6–46) |
| Practice nurse | 3 | 0 | 41 (31–57) | 8 (3–12) |
| Advanced care practitioner | 1 | 0 | 50 | 3 |
| Respiratory consultant | 1 | 1 | 55 | 31 |
| Respiratory registrar | 2 | 2 | 31 | 8 |
| Respiratory physiotherapist | 1 | 0 | 41 | 20 |
| Respiratory nurse | 1 | 0 | 48 | 13 |

"*You know, to have a spirometry done every year or two. And certainly, if there's a deterioration in symptoms, you know, doing another one at that point, just so that we can see clearly what is happening (. . .) So yeah, I quite like to be able to look back over several different readings and see how it's changed over a period of time*"–GP3

However, despite regular monitoring, patients found to have been misdiagnosed were not being identified for further review:

"*I'm not confident that it always the surgeries go back to reassess that. I think they just once they mentioned COPD on their read codes. . . . it's hard, isn't it to pick up where the diagnosis and when the diagnosis occurred*"–Respiratory Nurse1.

In some cases, despite patients being found to have normal spirometry at annual reviews by the nurse, the diagnosis would not be challenged as it had been assigned by a doctor. This was because GPs were perceived as the right HCPs to make the decision to investigate for possible COPD and then diagnose a patient as having COPD. Nurses perceived their role as one of monitoring disease and performing spirometry, but not to challenge the COPD diagnosis:

"*But we only do that if that comes from the GP. So, it's the GP that decides whether we do spiros, on which patients and when*"–Practice nurse1

Nonetheless, respondents did recognise that this was a potential area for better interprofessional working and change:

"*the nurse because of their expertise might actually be able to help the GP or the GP trainee, because we're a training practice, might help them to interpret the results (. . .) but in terms of putting their firm diagnosis on the record, it will be down to a doctor*"–GP3

Regular review of the patient's COPD diagnosis, alongside specialist support and education, was perceived as an intervention that would help reduce the prevalence of misdiagnosis within primary care:

"*Perhaps having a standard whereby you* [specialist] *review them once a year, at the very minimum. And when you review them, it could very well be that it comes to light then that the diagnosis wasn't made correctly*"–GP3

### Explanation 2—Challenges in differential diagnosis

Participants felt most COPD diagnoses were straightforward and could be made with clinical findings alone but would use spirometry to confirm their clinical assumptions and to meet QOF requirements:

"*if it's easy, you almost don't need this spirometry like you look at them and clinically, you make the diagnosis and you are almost certain, and you're like, "Yeah, you've got COPD" and you do spirometry to tick a box*"–GP4

However, participants did appreciate the availability of spirometry in primary care to assist with the diagnostic process when there was diagnostic uncertainty:

*"I think, it's the history that raises it, and then the spirometry might clinch it. But it's the history that's the starting point. But in terms of onus, I think, I think if you end up with spirometry and it's inconclusive, or it's not what you expect. Then you need to start thinking about other things I guess and getting help"–GP5*

Participants focused on their difficulty differentiating COPD from asthma despite making use of spirometry in primary care:

*"There are patients that have a mixed picture. In terms of when you look at his spirometry, it seems like they've got a bit of asthma and COPD as well. And I think it can be difficult for a GP that's not experienced . . . to hedge their bets and say it's COPD, and it could very well be that they're wrong, maybe COPD and asthma. I think it's unlikely that they will diagnose them with asthma"–GP3*

Participants perceived primary care as the appropriate environment for diagnosing patients with COPD, due to it being cost effective and easier for patients to access. However, participants appreciated that although primary care clinicians can diagnose the majority of patients, specialist involvement through MDT meetings would help improve confidence in the diagnoses being made in primary care:

*"So mentally, I think I think it's nice to have a specialist who can focus on it. And then we can have that discussion its useful to have a with our situation healthcare assistant who is very interested in COPD and chest medicine, and then GP with ongoing kind of interest and then having the specialist so we kind of bounce the ideas that description, that is what a multidisciplinary team is. And I think I think the days of having have one I'm the GP and I made a diagnoses, it doesn't hold you know . . . we should discuss it in a multidisciplinary team kind of way"–GP7*

## Discussion

### Main findings

Despite advancements in spirometry training and experience, reluctance to review historical COPD misdiagnoses appeared to be the leading cause of a persistent prevalence of COPD misdiagnosis in primary care. The findings from this study indicated that although difficulties interpreting spirometry were identified as the cause of misdiagnosis in the majority of cases, this was a simplistic view. Qualitative exploration revealed that primary care nurses were capable of interpreting spirometry, however, they were reluctant to challenge historical COPD misdiagnoses, probably due to ongoing hierarchies in the health professions [20]. However, the chronic management of COPD in primary care is often nurse led, with GP involvement during acute exacerbations. As a result, a cohort of historically misdiagnosed patients has persisted. Patients were likely to have initially been misdiagnosed 5–10 years prior to this study and due to differences in acceptable diagnostic procedure at that time, patients were labelled with COPD based on clinical findings alone.

In addition to reluctance to challenge historical misdiagnoses, this study also identified difficulty differentiating COPD and asthma as another key cause for misdiagnosis. Primary care HCPs had difficulty differentiating between COPD and asthma due to similarities in symptoms and concerns regarding Asthma-COPD Overlap Syndrome (ACOS) leading to uncertainty regarding treatment. As a result, due to diagnostic uncertainty clinicians perceived

treating for both asthma and COPD was the safest path, and so patients were labelled with both asthma and COPD.

## Contribution to literature

Existing literature focuses on the inadequacies of spirometry use and interpretation in primary care as the cause for COPD misdiagnosis [12]. The Welsh COPD audit identified a significant lack of spirometry recording in primary care but also identified that 26% of those with spirometry results recorded had results incompatible with a COPD diagnosis [21]. An audit of primary care in Hampshire reported over a 3 year period, 12% of patients had spirometry recorded that was inconsistent with COPD [22]. In both cases the authors suggested improvement in spirometry training were needed to address the issue of misdiagnosis in primary care. The quantitative findings from this study support the findings from the current literature, however, this is the first qualitative study exploring the causes of COPD misdiagnosis in primary care. This study contributes to the existing literature the concept of reluctance amongst primary care clinicians to challenge historical misdiagnoses of COPD as an important factor leading to persistent misdiagnosis of patients.

The perceived difficulty of differentiating between COPD and asthma amongst primary care clinicians has been demonstrated previously by Akindele et al. [23]. They explored, through qualitative interviews, challenges met when diagnosing asthma in primary care and found that clinicians had difficulty differentiating asthma from COPD [23]. The same perception of symptom similarity and difficulty differentiating asthma and COPD was found in this study. Diagnostic uncertainty between asthma and COPD amongst primary care HCPs resulted in patients having concomitant asthma and COPD diagnoses, this was similar to the findings in current primary care literature [24, 25]. This study corroborates existing evidence indicating that there is difficulty amongst primary care clinicians in differentiating between COPD and asthma. However, the study adds to current literature the potential of integrated care as an intervention to assist in reducing COPD misdiagnosis in primary care.

## Limitations

The quantitative data for this study was limited in the intervention arm to patients that had attended for their annual COPD review and agreed to be seen by a specialist as part of a trial. This limited the number of patients having diagnostic review directly completed and potentially underestimated the causes and prevalence of COPD misdiagnosis in the intervention practices. Whereas at the control sites all patients with a pre-existing COPD diagnosis underwent a specialist diagnostic review, albeit virtually, which prevented assessment to determine their underlying pathology. As with most qualitative studies, interview responses may have been influenced by the interviewer [26]. In this study the patients had been interviewed by a respiratory specialist and as such they may have altered their responses to questions knowing the interviewer was a specialist clinician. HCP interviews were divided equally between a respiratory specialist and a medical student. The transcripts from both interviewers were reviewed in a workshop with a non-clinical, qualitative researcher and it was judged that the responses did not vary between the two interviewers.

## Implications for practice

Primary care is the optimal setting for diagnosing patients with COPD, however, specialist support is needed for complex cases that are difficult to diagnose and support GPs when there is diagnostic uncertainty. Specialist support through "virtual" or "real" clinics would assist in

reducing misdiagnosis, however, its cost-effectiveness and acceptability needs to be assessed, this being part of the aim of the parent study (INTEGR-COPD).

Historically patients had been diagnosed with COPD without spirometric confirmation of irreversible airway obstruction due to standards of that time not requiring it. Due to historical poor diagnosing standards a cohort of misdiagnosed patients has emerged and remain misdiagnosed despite annual spirometry as part of COPD monitoring. Annual COPD monitoring is usually completed by practice nurses, who are experienced and capable of reading spirometry results, with minimal GP input. However, due to established job roles and boundaries, nurses often do not make or review COPD diagnoses. To reduce the prevalence of COPD misdiagnosis in primary care practice nurses need to be supported and encouraged to review and challenge COPD misdiagnoses when spirometry is not compatible with the diagnosis. Throughout the COVID19 pandemic spirometry ceased in primary care and patients have been diagnosed on the basis of clinical findings alone [27]. As a result, there is the potential for a new cohort of misdiagnosed patients and without encouragement to review their diagnoses, at annual monitoring review, they risk remaining misdiagnosed until reviewed by a specialist in secondary care or a primary care diagnostic hub [28, 29]. Therefore, the retraining of primary care clinicians to be confident and competent with spirometry needs to be a priority. Integrating specialists into primary care can facilitate on-the-job learning through the sharing of knowledge and expertise and potentially reverse the impact of the COVID19 pandemic on COPD misdiagnosis.

Future interventions to reduce COPD misdiagnosis in primary care may also include Artificial Intelligence (AI) [30]. Studies are currently ongoing to establish the feasibility and effectiveness of AI spirometry software to aid in the diagnosis of COPD. The full impact of AI spirometry is not yet fully understood, however, it is likely to play important role in primary care COPD diagnosis in the future.

## Conclusion

Misdiagnosis of COPD in primary care is likely to have occurred at a time when spirometry was rarely used, however, due to COPD diagnoses not being reviewed or challenged when subsequent spirometry is incompatible, a cohort of misdiagnosed patients has emerged in primary care. Newly misdiagnosed patients are likely to be misdiagnosed due to difficulty differentiating COPD from asthma, however, are unlikely to be disadvantaged through lack of treatment as they are likely to receive treatment for both COPD and Asthma. In order to reduce the prevalence of COPD misdiagnosis in primary care we need greater specialist involvement and support nurses to challenge historical COPD misdiagnoses.

## Supporting information

**S1 File. Topic guide.**
(DOCX)

**S2 File. Coding framework.**
(DOCX)

**S3 File. Category summaries.**
(DOCX)

**S4 File. Thematic map.**
(DOCX)

## Acknowledgments

We thank the patients and healthcare professionals of East Birmingham who participated in the INTEGR COPD study and the administrative staff at participating GP practices and Aisha Butt for their logistical support.

## Author Contributions

**Conceptualization:** Ketan Patel.

**Data curation:** Ketan Patel, Daniel J. Smith, Christopher C. Huntley, Sunita D. Channa.

**Formal analysis:** Ketan Patel, Daniel J. Smith, Christopher C. Huntley.

**Funding acquisition:** Alice M. Turner.

**Investigation:** Ketan Patel, Daniel J. Smith, Sunita D. Channa, Nicola Gale, Alice M. Turner.

**Methodology:** Ketan Patel, Nicola Gale, Alice M. Turner.

**Project administration:** Anita Pye.

**Supervision:** Sunita D. Channa, Anita Pye, Andrew P. Dickens, Nicola Gale, Alice M. Turner.

**Writing – original draft:** Ketan Patel.

**Writing – review & editing:** Ketan Patel, Daniel J. Smith, Christopher C. Huntley, Sunita D. Channa, Anita Pye, Andrew P. Dickens, Nicola Gale, Alice M. Turner.

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
