## [Decision Letter · Decision Letter 0]

21 Aug 2023

PONE-D-23-17867Exploring the causes of COPD misdiagnosis in Primary Care:  A mixed methods studyPLOS ONE

Dear Dr. Patel,

Thank you for submitting your manuscript to PLOS ONE. After careful consideration, we feel that it has merit but does not fully meet PLOS ONE’s publication criteria as it currently stands. Therefore, we invite you to submit a revised version of the manuscript that addresses the points raised during the review process.

We look forward to receiving your revised manuscript.

Kind regards,

Pisirai Ndarukwa, Ph.D.

Academic Editor

PLOS ONE

“I have read the journal's policy and the authors of this manuscript have the following competing interests: AMT and KP were supported financially in relation to this study through a non-commercial grant from AstraZeneca. AMT has received grants not in relation to this study from Chiesi, NIHR, CSL Behring and Grifols Biotherapeutics.

DJS, CCH, SDC, AP, APD and NG declare no competing interests exist.”

Additional Editor Comments:

I would like to inform you that the number of the reviewers required for this paper was not met as most declined and only one accepted and has concluded the review process. The comment was that there should be a major revision. I concur with the reviewer and accept his valuable comments.

Regards

Reviewers' comments:

Reviewer's Responses to Questions

**Comments to the Author**

1. Is the manuscript technically sound, and do the data support the conclusions?

Reviewer #1: Yes

2. Has the statistical analysis been performed appropriately and rigorously? 

Reviewer #1: Yes

3. Have the authors made all data underlying the findings in their manuscript fully available?

Reviewer #1: No

4. Is the manuscript presented in an intelligible fashion and written in standard English?

Reviewer #1: Yes

5. Review Comments to the Author

Reviewer #1: This manuscript explores a very important and timely issue, namely the misdiagnosis of people with COPD within primary care. The report is well-written, the authors present the aims of the study clearly and within the context of the larger trial. The methods section is missing a few important details and I believe the authors would benefit from using a reporting checklist such as COREQ to ensure everything is covered (e.g. both interviewer’s characteristics). I have added a few questions/comments for your consideration:

1. Was the PROGRESS_PLUS or any other tool used to support maximum variation sampling?

2. How was theoretical saturation determined?

3. Were any patient and public involvement members involved in the development of the interview schedules or study design?

4. Do the authors have access to any socioeconomic information that could be included in table 1?

5. Please include a participant flow diagram in the results section. How many HCPs and patients were approached to be included in the qualitative component. How many refused and why?

6. The implications for practise are well covered, but I wonder if implications for training future HCPs might also be mentioned. It might also be necessary to mention the development of potential AI solutions to facilitate accurate diagnosis.

6. PLOS authors have the option to publish the peer review history of their article (what does this mean?). If published, this will include your full peer review and any attached files.

Reviewer #1: No

---

## [Author Response · Author response to Decision Letter 0]

6 Nov 2023

Please find below our responses to comments from the Editor:

1. As recommended, we have edited our manuscript to meet the journal’s style requirements. 

2. We have amended the manuscript to include that all participants gave informed written consent.

3. Our updated competing interests includes the statement “This does not alter our adherence to PLOS ONE policies on sharing data and materials.”.

4. We have included the minimal data set as a supplement in supplement files 2, 3 and 4.

5. We have amended the manuscript to include the ethics statement in the Methods section.

Please find below our responses to the Reviewers comments:

1. We have completed and attached the COREQ checklist as recommended.

2. Purposeful sampling was conducted through assessing the demographics of potential participants and identifying participants to approach to ensure heterogeneity. No software or tools were used to support this process.

3. Theoretical saturation was determined based on the review of ideas emerging from consecutive interviews, once the lead researcher felt no new ideas were emerging the possibility of saturation was discussed within the research team and once all researchers in the team were in agreement that saturation had been reached further recruitment ceased. The research team consisted of 3 clinical researchers, 3 non-clinical researchers and a medical student.

4. Patient representatives were involved through a trial steering committee, they gave their thoughts on the study design and development of the topic guide, the manuscript has been amended in the Methods section to include this aspect.

5. Socioeconomic data has been added to table 1 as recommended. 

6. We have opted against illustrating recruitment to the qualitative stage of the study with a flow diagram as the same information can be expressed through prose. The Results section has been amended to include number of patients and healthcare professionals approached, we did not record the reason given for declining to participate.

7. The implications for practice has been amended to include implications of AI and implications on training as recommended.

---

## [Decision Letter · Decision Letter 1]

25 Jan 2024

Exploring the causes of COPD misdiagnosis in Primary Care:  A mixed methods study

PONE-D-23-17867R1

Dear Dr. Patel,

We’re pleased to inform you that your manuscript has been judged scientifically suitable for publication and will be formally accepted for publication once it meets all outstanding technical requirements.

Kind regards,

Pisirai Ndarukwa, Ph.D.

Academic Editor

PLOS ONE

Additional Editor Comments (optional):

Reviewers' comments:

Reviewer's Responses to Questions

**Comments to the Author**

1. If the authors have adequately addressed your comments raised in a previous round of review and you feel that this manuscript is now acceptable for publication, you may indicate that here to bypass the “Comments to the Author” section, enter your conflict of interest statement in the “Confidential to Editor” section, and submit your "Accept" recommendation.

Reviewer #1: All comments have been addressed

2. Is the manuscript technically sound, and do the data support the conclusions?

Reviewer #1: Yes

3. Has the statistical analysis been performed appropriately and rigorously? 

Reviewer #1: N/A

4. Have the authors made all data underlying the findings in their manuscript fully available?

Reviewer #1: Yes

5. Is the manuscript presented in an intelligible fashion and written in standard English?

Reviewer #1: Yes

6. Review Comments to the Author

Reviewer #1: (No Response)

7. PLOS authors have the option to publish the peer review history of their article (what does this mean?). If published, this will include your full peer review and any attached files.

Reviewer #1: **Yes: **Samantha L. Harrison

---

## [Editor Report · Acceptance letter]

26 Feb 2024

PONE-D-23-17867R1 

PLOS ONE

Dear Dr. Patel, 

I'm pleased to inform you that your manuscript has been deemed suitable for publication in PLOS ONE. Congratulations! Your manuscript is now being handed over to our production team.

Kind regards, 

on behalf of

Prof Pisirai Ndarukwa 

Academic Editor

PLOS ONE